# Customized Textile Capacitive Insole Sensor for Center of Pressure Analysis

**DOI:** 10.3390/s22239390

**Published:** 2022-12-01

**Authors:** Jong-Gab Ho, Young Kim, Se-Dong Min

**Affiliations:** 1Department of Software Convergence, Soonchunhyang University, Asan 31538, Republic of Korea; 2Department of Medical IT Engineering, Soonchunhyang University, Asan 31538, Republic of Korea

**Keywords:** gait, insole, textile capacitive pressure sensor, wearable sensor, center of pressure

## Abstract

Center of pressure refers to the centroid of the ground reaction force vector detected underneath the walking foot, which is a summary measure representing body segment movements during human locomotion. In this study, we developed a cost-effective, lightweight insole-type textile capacitive sensor (I-TCPs) to analyze plantar pressure (PP) distribution and center of pressure (COP) trajectory. To test the accuracy of I-TCPs, the measured pressure data was compared with that of F-scan. The sensor performance test was divided into a static baseline test and a dynamic gait experiment, both at two different gait speeds self-selected by the subjects. Static gait results showed that I-TCPs were capable of recognizing PP segments at different gait speeds. Dynamic gait results showed an average RMSE of 1.29 ± 0.47 mm in COPx (mediolateral shift) and 12.55 ± 5.08 mm in COPy (anteroposterior shift) at a comfortable gait speed. The COP correlation between I-TCPs and F-scan was 0.54 ± 0.09 in COPx and 0.92 ± 0.04 in COPy in comfortable gait speed conditions, in which COPy values presented a stronger correlation. RMSE and correlation in fast gait speed conditions also showed similar results. The findings of this study can be the basis for future research, including rehabilitation engineering, developing ICT devices, and creating smart wearable sensors to improve quality of life for patients and healthy individuals.

## 1. Introduction

Pedobarography is the study of the pressure created between the plantar surface of the foot and the supporting surface during dynamic loading [1]. Researchers, clinicians, and engineers use pedobarography as a diagnostic tool to investigate foot-ground interaction, foot posture, balance functions, and to screen for pressure-related abnormalities in healthy individuals that are likely to lead to pathologies [2]. Information on plantar pressure (PP) is known to be useful in analyzing and predicting gait disorders in patients with stroke, diabetes, orthopedic surgeries, or sports injuries in the lower extremity [3]. In addition, it is used for research on personal identification [4,5,6,7] and safety monitoring [8,9] using PP. Accordingly, the demand for the advancement of various wearable devices is continuously growing, and smart sensors for human locomotion and health state monitoring are gaining in importance.

Center of pressure (COP) refers to the centroid of the ground reaction force (GRF) vector detected underneath the foot, which is a summary measure representing body segment movements during locomotion [10]. Tracking COP is one of the basic parameters used for human gait analysis, as the changes in PP are meaningful in evaluating the stability and function of the foot and ankle, which play a key role in anticipatory movements [11]. The dynamic properties of COP excursion, also noted as ‘gait line’, are used in health state monitoring, gait characteristic recognition, and senile dementia prediction [12]. Postural control regulates the ability to maintain an upright stance and is essential in performing activities of daily living [13].

The standard and most accurate method to measure COP is to use a force plate embedded in the ground to capture the instantaneous ground reaction force in a still stance. Such static foot measures are used as inferences for dynamic PP measures [14]. However, this lab-based environment is not identical to the situations and terrains in real life and thus can restrict or alter the subjects’ natural gait patterns [15]. In the last decade, numerous researchers have developed and studied various types of wearable sensors; some were synced with smartphone applications to monitor and analyze human gait in daily-life conditions, finding ways to measure dynamic PP from within the footwear [12,15,16].

Recently, smart insoles and sensorized shoes have been noted to be gaining popularity due to their portability and availability for outdoor activity monitoring. These human-centered technologies are growing rapidly along with people’s rising social standards and quality of life. The latest sensorized shoes can be custom equipped with other types of sensors, including an accelerometer, gyroscope, and bend sensor, and can measure GRF and COP with reliable accuracy [17]. However, the cost is still relatively high, the sensors are heavy and rigid, and the analysis is complicated and time-consuming.

Pressure-detecting sensors can be made with resistive, capacitive, or piezoelectric transducers. Recent interest in wearable sensors is leading to simple, daily-usable devices that are capable of logging objective gait data indexes for real-time analysis. Most of the previous ‘smart-design’ insoles seemed to be based on piezo-resistive force sensors [18,19,20,21,22]. Compared to force-sensitive resistors (FSR) or piezo-resistive force sensors, capacitive pressure sensors (CPS) are reported to have the advantages of a simple structure, low power consumption, and signal anti-interference. Capacitive sensors feature high levels of performance in terms of stability, repeatability, and linearity of conductivity with applied force [12,23]. CPS also possesses a wide detection range, which enables the fabrication of various forms and sizes. This makes CPS an optimum option for customization, which is most needed in wearable sensors, especially for insoles.

Previous study results confirmed that repeatability is one of the essential factors in insole sensors, considering an average of 6000 step counts per day for healthy individuals. Flexible and soft materials, such as textile CPS, are lightweight, customizable, and durable; thus, they are preferred particularly for insole sensors since they are worn with shoes to walk in [12,16]. These features of the CPS make it possible to measure dynamic pressure changes while the wearer walks or runs, making real-time analyses possible. In our previous study, we developed a CPS based on conductive textile and conducted comparative analysis studies on step count and stride time at normal walking speed [24].

## 2. Methods

### 2.1. Plantar Pressure Measurement System

To measure and analyze plantar pressure during static and dynamic gait, we developed an insole-type textile capacitive sensor (I-TCPs) using the conductive textiles of W-290-PCN (A-jin Electron, Busan, Republic of Korea). The specifications of W-290-PCN are described in Table 1. The design image of the actual sensor placements in I-TCPs is shown in Figure 1. The thickness of the insole sensors is 1.7 mm.

The I-TCPs presented in this study are a modified version of our previously developed insole sensor [24]. For I-TCP, a textile pressure sensor (W-290-PCN) with a size of 3 × 3 cm was used, and its thickness was 1 cm. To test sensor performance, compressive pressure variations from 0 to 110 kPa [25] and repetitions of 5000 times with a force of 450 kPa were used [8]. Based on the findings from our previous studies, 5 capacitive sensors were placed in the forefoot area, 2 in the lateral midfoot, and 3 in the rearfoot area for more accurate detection of PP during dynamic gait (Figure 1). In normal locomotion, the heel, metatarsals, and phalanges support most of the body weight and adjust the body balance [26]. These anatomical structures function as the medium to absorb and deliver the contact force derived from the foot during gait in the form of ascending neurophysiological information [27]. To capture the characteristic plantar pressure during gait, a total of 10 capacitive sensors in an insole were placed over the most representative anatomical locations of the walking foot [17,24].

Figure 2 shows the composition of the textile capacitive sensor and measurement board that collects and processes the data detected by the insole sensor [24]. Data collected from the sensor were processed through a capacitance to digital converter (CDC) on the measurement board. The detection of plantar pressure, collection, processing, and saving of the data were all performed through wireless communication in the monitoring application system before the analysis procedure (Figure 3).

In order to extract COP data collected from I-TCPs, x-coordinates (mediolateral shifting of COP) and y-coordinates (anteroposterior shifting of COP) for each sensor were specified and calculated as shown below in Figure 4 [19]. Figure 4 presents the detailed coordinate information for each sensor channel of female I-TCPs in a 235 mm insole with 78 mm width.

### 2.2. Performance Test of the Plantar Pressure Measurement System

To test the performance and accuracy level of our developed sensor (I-TCPs), the pressure data collected by I-TCPs were compared with that of the F-scan (Tekscan, Boston, MA, USA), a commercial pressure sensor with high reliability and validity. For simultaneous data collection and direct comparison, I-TCPs were placed over the F-scan sensor inside the sneakers provided for the subjects participating in this study [17,18]. Sensor performance tests were divided into two separate experiments, as described below.

#### 2.2.1. Plantar Pressure Segment Recognition (Baseline Test)

To examine the basic performance of I-TCPs, the plantar pressure (PP) recognition ability of the insole sensor was first tested in a static condition. As a baseline test, a male subject was to stand still in one designated area and then march in place for 2 min when a verbal cue was given. In order to test whether I-TCPs detect speed-associated PP segment differences, two different foot-stamping speeds were performed: first, simulating a comfortable normal gait speed (slow speed), and then simulating running (fast speed) in place for 2 min. Each test condition was performed three times with a 2 min break in between. The sampling rate was set at 100 Hz [28]. Stride time and cadence were calculated to extract COP. This test was performed to establish the baseline for the analysis of the main experiment.

#### 2.2.2. Dynamic Gait COP Measurement (Main Experiment)

For the main experiment of this study, a total of 4 female subjects participated (age: 23.0 ± 0.7 years, height: 158.8 ± 1.5 cm, weight: 49.5 ± 1.8 kg) (Table 2). The subjects were to walk on the treadmill for 3 min at self-selected comfortable and fast speeds. Table 2 below shows the actual comfortable and fast speeds selected by each subject.

For data acquisition, each subject performed the walking experiment for 2 min for each condition in a random order. A five-minute resting time was given in between tests to minimize the effect of the previous experiment. The sampling rate was set at 100 Hz.

### 2.3. Gait Signal Processing

For analysis of the collected gait signals, data processing was performed in the order shown below in Figure 5. First, preprocessing was performed, and then Heel Strike (HS) and Toe Off (TO) indicators were extracted. Then, the gait index was calculated based on HS and TO. All signal processing and analysis were performed by MATLAB R2021a.

To extract the gait indexes of HS and TO from the acquired gait signal, the DC component was first removed to set the reference point of the data to a value of zero for signal calibration. A low-pass filter (cut-off frequency of 3 Hz) was applied to remove noise in the high-frequency component, and the slope component value of the walking signal was calculated through the derivative filter. Data squared was then performed for amplification of positive values, and HS and TO indexes were extracted through the peak PP detection algorithm. The highest peak value of PP at HS was calculated by averaging the PP values from channels 9 and 10 (the heel area of the insole). For gait index analysis, the following formula was applied to extract stride time, cadence, and COP information.

### 2.4. Gait Index Extraction

#### 2.4.1. Stride Time, Stance and Swing Phase

As shown below in Figure 6, HS and TO were extracted from I-TCPs data. As HS refers to the initial heel contact at the beginning of each gait cycle, the blue line graph in Figure 6 shows the changes in peak PP collected from the heel area of the insole (peak average of ch. 9 and ch. 10). The orange line graph shows the changes in peak PP in the toe area (peak average of ch. 1 and ch. 2). Based on every heel strike, stride time was calculated for each gait cycle (Equation (1)). According to the PP data detected in the forefoot, midfoot, and heel areas, the stance phase and swing phase were defined (Figure 6).
(1)$${\mathrm{Stride}\;\mathrm{time}(s)=\mathrm{HS}}_{i+1}{\;-\;\mathrm{HS}}_{i}$$

#### 2.4.2. Cadence

To calculate cadence, the total stride count was derived by counting the number of HS peak points (Figure 6). Since each gait was measured for 2 min, the time variable in Equation (2) was 120 s. The derived value was then multiplied by 60 s to get the step counts per minute.
(2)$$\mathrm{Cadence}\left(\frac{\mathrm{steps}}{\min}\right)=\frac{\mathrm{Total}\;\mathrm{Stride}\;\mathrm{count}}{\mathrm{Times}}\times 60\;s$$

#### 2.4.3. Center of Pressure

For COP analysis, the x-coordinate (mediolateral shift) and y-coordinate (anteroposterior shift) values were calculated by Equations (3) and (4) based on the PP data collected by I-TCPs. The x-coordinate and y-coordinate were calculated using the coordinate information from the insole sensor (Figure 4). The direction of COP_(x,y)_ was used for COP trajectory analysis. Figure 7 below shows the process of COP extraction based on PP obtained from I-TCPs. An average of five samples was defined as one frame, which makes 100 data points per second (100 Hz), displaying 20 frames per second:(3)$${\mathrm{CoP}}_{x}=\frac{\sum_{i=1}^{n}{\mathrm{Pr}}_{i}{\;\times \;X}_{i}}{\sum_{i=1}^{n}{\mathrm{Pr}}_{i}},$$
(4)$${\mathrm{CoP}}_{y}=\frac{\sum_{i=1}^{n}{\mathrm{Pr}}_{i}{\;\times \;Y}_{i}}{\sum_{i=1}^{n}{\mathrm{Pr}}_{i}},$$

in the equations above, Pr_i_ refers to the pressure value detected by each sensor channel, X_i_ refers to the x-coordinate, and Y_i_ refers to the y-coordinate value of the sensor.

### 2.5. Data Anlaysis

For performance comparison between I-TCPs and F-scan, root mean square error (RMSE) and correlation coefficient (ρ) were calculated by using the Equations (5) and (6) as shown below:(5)$$\mathrm{RMSE}=\sqrt{\frac{\sum_{i=1}^{N}{{(x}_{i}-\hat{x_{i}})}^{2}{+(y}_{i}-\hat{y_{i}}{)}^{2}}{N}},$$
(6)$$\mathrm{Correlation}\;\mathrm{coefficient}=\frac{\sum_{i}^{n}{(X}_{i}-\overset{-}{X}){(Y}_{i}-\overset{-}{Y})}{\sqrt{\sum_{i}^{n}{{(X}_{i}-\overset{-}{X})}^{2}}\sqrt{\sum_{i}^{n}{{(Y}_{i}-\overset{-}{Y})}^{2}}}.$$

## 3. Results of the Plantar Pressure Measurement System Performance Test

### 3.1. Plantar Pressure Segment Recognition

Figure 8 shows PP segment recognition results from the baseline test. Each frame represents an average of five samples of PP data detected by I-TCPs. During static foot stamping at the self-selected comfortable walking speed, PP distribution was mainly concentrated in the rearfoot and heel areas (Figure 8a). However, in the running-simulated speed condition, PP distribution was noticeably localized in the forefoot area (Figure 8b). Baseline test results showed the capability of I-TCPs in recognizing PP segments and body weight shifting in the foot during different movements.

From the baseline test, performance comparison results between I-TCPs and F-scan in static gait showed a 100% agreement in cadence (steps/min) with a correlation coefficient of r = 1 (Table 3). The cadence in the comfortable walking speed condition was measured as 50.00 ± 2.29 by I-TCPs and 50.00 ± 2.29 by F-scan. The cadence in the running-simulated speed condition was measured as 77.67 ± 0.29 by I-TCPs and 77.67 ± 0.29 by F-scan.

### 3.2. Dynamic Gait COP Measurement

A total of 355 dynamic gait strides in the self-selected comfortable speed condition and 590 strides in the fast speed condition, which were collected from four subjects, were used for analysis. The x-coordinate (mediolateral movement) and y-coordinate (anteroposterior movement) COP values from the two gait speeds (comfortable and fast) were analyzed for comparison of RMSE and correlation between I-TCPs and F-scan.

As a result, the average RMSE of COPx at a comfortable gait speed was 1.29 ± 0.47 mm and that of COPy was 12.55 ± 5.08 mm, showing better accuracy with lower error values in COPx. The results of the COP measurement correlation between I-TCPs and F-scan were 0.54 ± 0.09 in COPx and 0.92 ± 0.04 in COPy in a comfortable gait speed condition, in which COPy values presented a stronger correlation (Table 4).

The average RMSE of COPx was 1.34 ± 0.47 mm and that of COPy was 11.26 ± 3.89 mm, showing better accuracy with lower error values in COPx. The results of the COP measurement correlation between I-TCPs and F-scan were 0.49 ± 0.22 in COPx and 0.94 ± 0.03 in COPy in the fast gait speed condition, in which COPy values presented a stronger correlation (Table 5).

## 4. Discussion

In this study, an in-shoe pressure-measuring textile capacitive pressure sensor (I-TCPs) capable of PP segmentation and COP analyses was proposed. I-TCPs were built with only 10 capacitive sensors for each side of the foot, making them cost-effective yet aiming for accurate measurement by placing the sensors on the most anatomically representative spots in the walking foot. We believe that developing simple and customizable, user-friendly wearable insole sensors is meaningful in the field of preventive healthcare because how a person walks represents whole-body health status and postural control, which are directly related to quality of life.

To test the performance and accuracy level of our developed sensor (I-TCPs), major gait indices, including stride time, cadence, and COP (x- and y-coordinates), in the stance and swing phases of a gait cycle were analyzed at two different gait speeds. The reason for selecting a comfortable gait speed in static and dynamic conditions was to examine whether I-TCPs are feasible for real-life application during activities of daily living (ADL).

For the basic performance of I-TCPs, PP segment recognition ability was first tested in a static standing condition, and since this was a baseline test, single subject data at two different foot-stamping speeds were collected and analyzed. PP segment results from I-TCPs agreed 100% with those of a commercial sensor; PP distribution was concentrated in the rearfoot area at walking speed and in the forefoot area at running speed. The location of the COP during quiet standing is known to change as we adjust our muscles to maintain balance. Two common measurements used to analyze the quality of postural control are the amount and direction of movement. COP is used to control the center of mass (COM); when COM moves forward, body weight shifts towards the toes (forefoot) [13]. This shift was also found in the running-simulated condition of the static gait test in our study.

The results of our study showed higher accuracy with lower error values in COPx (1.29 ± 0.47 mm in slower gait, 1.34 ± 0.47 mm in faster gait) compared to COPy (12.55 ± 5.08 mm in slower gait, 11.26 ± 3.89 mm in faster gait). Similar findings were reported in the studies by Zhang et al. [22] and Morin et al. [29]. According to their study results, locating the sensors at the top and bottom edges of the insole may be a method to minimize the error rate in COPy.

The results of the correlation analysis between I-TCPs and F-scan showed that the COPy values had a stronger correlation. The y-coordinate is the anteroposterior COP shift, which represents the forward progression during gait. Although I-TCPs had 10 sensor channels, the PP distribution in the overall foot area and COP trajectory well matched the F-scan data results.

As shown in Table 6, previous studies also reported fewer variant factors in the mediolateral COP shifts during gait with higher error sensitivity in the x-coordinate [19]. The findings of our study were similar to the results of the study by Choi et al. [18], where the COP_(x,y)_ was derived from the FSR sensor using the weighted mean approach (WMA) method at a walking speed of 2 km/h, and a comparative study was performed with the F-scan system [18]. The WMA method used by Choi et al. was the same method applied to the capacitive sensor in this study. The experiment was performed at speeds of 1 km/h, 2 km/h, squatting, and sitting to stand. Among them, the RMSE result of FSR sensor data at 2 km/h was 19.01 mm, which was higher than the RMSE result of 12.62 mm in the comfortable speed condition of this study. The smaller the value of RMSE, the smaller the COP measurement error with the reference instrument. Although the number of sensors used in our study was higher (ten channels) compared to the six channels used by Choi et al., capacitive sensors were found to have better performance in detecting PP than resistive FSR sensors. In addition, the correlation coefficient results of COP in this study (COPx = 0.95, COPy = 0.54) had similar results to those of the study by Choi et al. (COPx = 0.97, COPy = 0.55) and were relatively similar [18].

The revised and upgraded version of our developed insole sensor presented the possibility of PP segmentation and COP analysis for static and dynamic gait and showed similar performance when compared with commercial systems. Wearable capacitive insole sensors such as I-TCPs can overcome the high cost and space constraints for measuring gait data, which are the often-discussed limitations of existing commercial PP measurement equipment.

The findings of this study can be the basis for future research, including rehabilitation engineering, developing ICT devices, and using smart wearable sensors to improve quality of life not only for patients under medical care but also for healthy individuals.

## 5. Conclusions and Future Work

In this study, a simple and easy-to-wear textile capacitive pressure insole sensor (I-TCPs) capable of PP segmentation and COP analyses was proposed. Future work of related studies is expected to meet higher quality standards and demands of our society, as it is cost-effective while demonstrating accurate measurement capability and agreeing 100% with a reputable commercial gait measurement system.

In our next study, various types of insole sensors will be manufactured considering the optimal sensor locations for various foot types and ages of the users. Additionally, additional research will be conducted to extract gait asymmetry, phase coordination index, and gait segmentation, which are the key gait indicators for monitoring health status. Based on this, we plan to expand the study of gait monitoring systems to derive more health-related gait indexes with insole-type capacitive sensors.

## Figures and Tables

**Figure 1 sensors-22-09390-f001:**
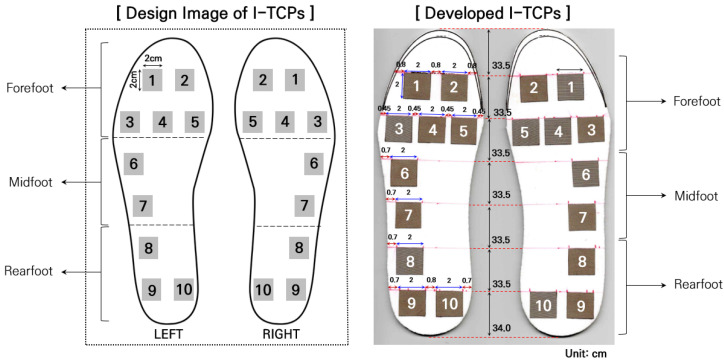
Design of insole-type textile capacitive sensors (I-TCPs).

**Figure 2 sensors-22-09390-f002:**
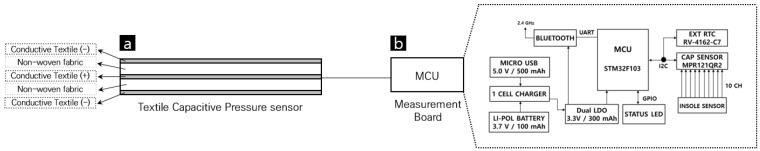
Structure of (**a**) textile capacitive sensor and (**b**) measurement board.

**Figure 3 sensors-22-09390-f003:**
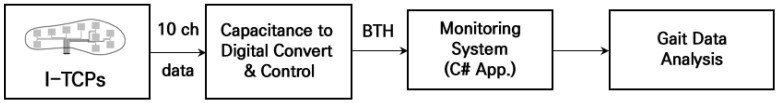
Gait data acquisition process.

**Figure 4 sensors-22-09390-f004:**
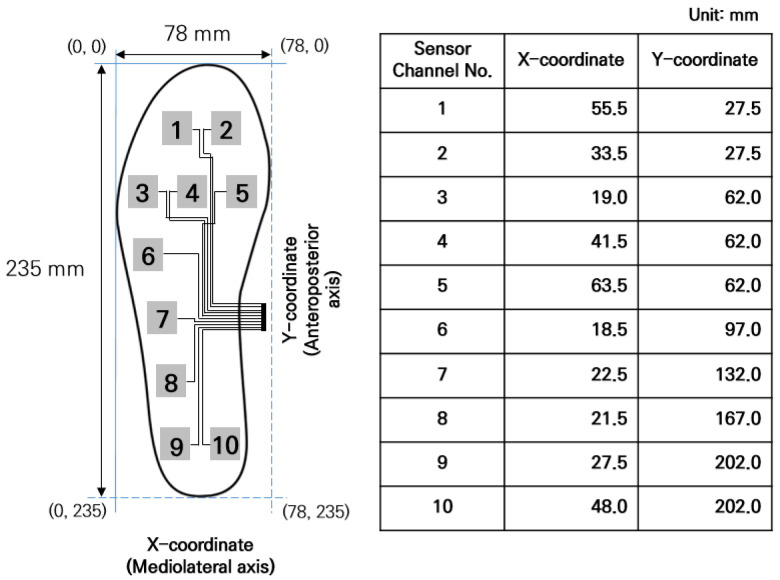
Coordinate information detected by each sensor channel of I-TCPs.

**Figure 5 sensors-22-09390-f005:**
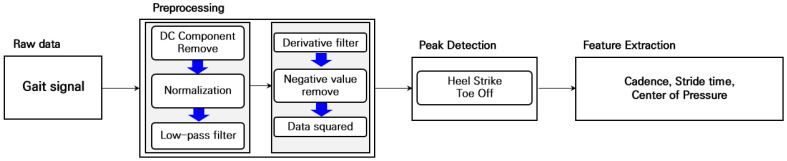
Flowchart of gait data preprocessing and feature extraction algorithm.

**Figure 6 sensors-22-09390-f006:**
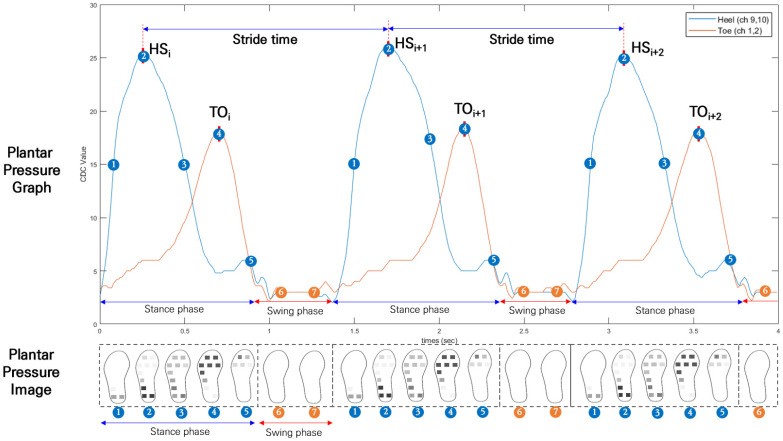
Extraction of stride time, stance phase, and swing phase from I-TCP data.

**Figure 7 sensors-22-09390-f007:**
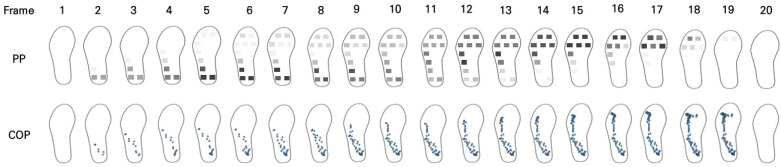
COP extraction process using I-TCPs.

**Figure 8 sensors-22-09390-f008:**
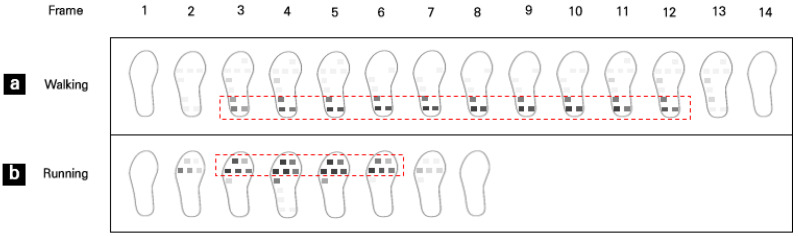
PP segment recognition results of foot stamp during simulated (**a**) walking, (**b**) running.

**Table 1 sensors-22-09390-t001:** Specifications of W-290-PCN.

Parameters	W-290-PCN
Plating type	Nickel, Copper
Fabric type	Woven
Weight (g/m^2^)	81 ± 5
Thickness (mm)	0.1 ± 0.01
Breaking strength (kg/5 cm)	392 ± 10

**Table 2 sensors-22-09390-t002:** Anthropometric and general characteristics of the subjects.

Variables		Values				
		Subject 1	Subject 2	Subject 3	Subject 4	Ave ± SD
Gender (M/F)		Female	Female	Female	Female	-
Age (years)		24	23	23	22	23.00 ± 0.82
Height (cm)		157	159	161	158	158.75 ± 1.71
Weight (kg)		47	49	50	52	49.50 ± 2.08
Body Mass Index (kg/m^2^)		19	19	19	21	19.64 ± 0.80
Shoe size (mm)		230	235	235	230	
Gait speed (km/h)	Comfortable	2.50	2.50	2.30	2.20	2.38 ± 0.15
	Fast	5.00	5.00	5.00	5.00	5.00 ± 0.00

**Table 3 sensors-22-09390-t003:** Result of a static gait index comparison between I-TCPs and F-scan.

		I-TCPs	F-scan
Walking	Cadence (steps/min)	50.00 ± 2.29 *	50.00 ± 2.29 *
Running	Cadence (steps/min)	77.67 ± 0.29 *	77.67 ± 0.29 *

*: Ave ± SD.

**Table 4 sensors-22-09390-t004:** Results of RMSE and correlation between I-TCPs and F-scan at comfortable gait speeds.

Comfortable Speed						
Subject No.		1	2	3	4	Ave ± SD
RMSE (mm)	* COP_x_	1.07	1.56	1.77	0.74	1.29 ± 0.47
	** COP_y_	7.99	12.38	19.68	10.15	12.55 ± 5.08
	*** COP_(x,y)_	8.06	12.48	19.76	10.18	12.62 ± 5.10
Correlation Coefficient	* COP_x_	0.47	0.58	0.65	0.46	0.54 ± 0.09
	** COP_y_	0.97	0.98	0.92	0.94	0.95 ± 0.03
	*** COP_(x,y)_	0.98	0.95	0.98	0.97	0.97 ± 0.01

* COP_(x)_: x-coordinate of COP (mediolateral shift), ** COP_(y)_: y-coordinate of COP (anteroposterior shift), *** COP_(x,y)_: x and y coordinates of COP.

**Table 5 sensors-22-09390-t005:** RMSE results and the correlation between I-TCPs and F-scan in fast gait speed.

Fast Speed						
Subject No.		1	2	3	4	Ave ± SD
RMSE (mm)	* COP_x_	1.37	1.18	1.96	0.83	1.34 ± 0.47
	** COP_y_	11.01	13.16	5.93	14.92	11.26 ± 3.89
	*** COP_(x,y)_	11.09	13.21	6.25	14.94	11.37 ± 3.76
Correlation Coefficient	* COP_x_	0.77	0.47	0.50	0.23	0.49 ± 0.22
	** COP_y_	0.97	0.93	0.91	0.94	0.94 ± 0.03
	*** COP_(x,y)_	0.95	0.96	0.96	0.92	0.95 ± 0.02

* COP_(x)_: x-coordinate of COP (mediolateral shift), ** COP_(y)_: y-coordinate of COP (anteroposterior shift), *** COP_(x,y)_: x and y coordinates of COP.

**Table 6 sensors-22-09390-t006:** Comparison of COP analysis using various types of insole sensors.

Author	Sensor Type	Condition (Gait Speed)	RMSE (mm)		
			COP_x_	COP_y_	COP_(x,y)_
Choi et al. [18] (2018)	Force Sensitive Resistor (six channels)	Walking (1 km/h)	-	-	19.01
		Walking (2 km/h)	-	-	17.15
Zhang et al. [22] (2017)	Piezo-resistive (eight channels)	Walking (Comfortable)	4.10 ± 0.60	10.10 ± 2.90	-
		Running (Comfortable)	4.20 ± 0.50	10.90 ± 1.80	-
Morin et al. [29] (2022)	Capacitive (sixteen channels)	Walking (Comfortable)	4.20 ± 1.00	12.00 ± 3.00	-
		Running (Comfortable)	5.30 ± 2.00	16.00 ± 5.00	-
**Below is the results from this study for comparison with above references.**					
	Capacitive (ten channels)	Walking (2.38 km/h)	1.29 ± 0.47	12.55 ± 5.08	12.62 ± 5.10
		Walking (5 km/h)	1.34 ± 0.47	11.26 ± 3.89	11.37 ± 3.76

## Data Availability

Not applicable.

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
