# Peer review of "Customized Textile Capacitive Insole Sensor for Center of Pressure Analysis"

_sensors, 2022, doi:10.3390/s22239390_

Round 1
Reviewer 1 Report
Dear authors, I see following flaws of Your paper:
1. X and Y direction should be named/explained as a-p and m-l earlier, not in the half of the paper.
2. I will not call GRFs bio-signals, those are EMG, ECG, EEG
3. only 10 sensors and big ones (2x2cm) per foot result in very low resolution of the system. What is the advantage (besides price) of Your system? On the market, there exist much more advanced insoles for pedobarographic measurements
4. why the insoles are not perfectly symmetrical? (distribution of the sensors for the left and right insole is slightly different)
5. "A total of 10 capacitive sensors were placed over the most anatomically representative spots in the walking foot [17,24]" – what are dose spots according to authors and why they are so important - this part needs a deeper explanation. What with differences in foot geometry for different patients - what with measurement accuracy?
6.The way of testing the sensors influences the results: ” To test the performance and accuracy level of our developed sensor (I-TCPs), the pressure data collected by I-TCPs was compared with that of F-scan (Tekscan, Boston, USA), a commercial pressure sensor with high reliability and validity. For simultaneous data collection and direct comparison, I-TCPs was placed over F-scan sensor inside the sneakers provided for the subjects participating in this study. Sensor performance test was divided into two separate experiments as described below.”
This influences the pressure distribution - this part needs deeper explanation.
7. How thick and stiff is Your insole?
8. Based on the plantar pressure data (darker=higher pressure?) we can assume that the maximum is before TO – in minimum visible in plantar pressure graph. Explanation, better description is necessary
9. HS is a point, value or it is time – it is unclear, especially according to rule no 1.
10. what means "x" in eq. 3 and 4 - simple multiplication or cross product or something else?
11. What is the novelty of Your project?
12. You used only 4 volunteers to test the insoles and all of them were women.
Reviewer 2 Report
This manuscript reported a cost-effective, lightweight insole type textile capacitive sensor (I-TCPs) to analyze plantar pressure (PP) distribution and center of pressure (COP) trajectory, which is meaningful to rehabilitation engineering and development of ICT devices and wearing sensors. This is an interesting work that I would recommend for publication in Sensors after addressing the following:
1. The authors are suggested to test the signal anti-interference and repeatability of the I-TCPs, both of which are essential in practical applications, as the authors mentioned in Introduction.
2. The high performance of I-TCPs is not that convincing due to the lack of comparisons. In addition to the FSR sensor reported by Choi et al., the authors are suggested to draw a comparison on RMSE results and correlation coefficient results between this I-TCPs and other existing force-sensitive resistors and capacitive pressure sensors.
3. In addition, the Method and Discussion sections are not well organized. Some results and statements are mentioned repeatedly in both sections.
Reviewer 3 Report
The present manuscript is clearly described and with enough results to demonstrate the proposal for capacitive sensors. From my personal point of view, it's well-written and interesting for some biomedical analysis. I recommend the publication as it is.
Author Response
Thank you for making the time to review our paper. We appreciate your review comments.
Round 2
Reviewer 1 Report
Dear Authors, thank You for the reply. I accept them.
One remark: You added BMI index - it is a useless index, nevertheless I assume that this is the demand of the second reviewer. The better one is the body fat amount.
Reviewer 2 Report
The authors have solved my concerns and I have no further comments on this manuscript.